# Urinary Exosomes of Patients with Cystic Fibrosis Unravel CFTR-Related Renal Disease

**DOI:** 10.3390/ijms21186625

**Published:** 2020-09-10

**Authors:** Sebastien Gauthier, Iwona Pranke, Vincent Jung, Loredana Martignetti, Véronique Stoven, Thao Nguyen-Khoa, Michaela Semeraro, Alexandre Hinzpeter, Aleksander Edelman, Ida Chiara Guerrera, Isabelle Sermet-Gaudelus

**Affiliations:** 1Institut Necker Enfants Malades, INSERM U1151, 75015 Paris, France; sa_gauthier@hotmail.com (S.G.); iwona.pranke@inserm.fr (I.P.); vincent.jung@inserm.fr (V.J.); thao.nguyen-khoa@aphp.fr (T.N.-K.); alexandre.hinzpeter@inserm.fr (A.H.); aleksander.edelman@inserm.fr (A.E.); chiara.guerrera@inserm.fr (I.C.G.); 2Proteomics Platform Necker, Structure Fédérative de Recherche Necker, 75015 Paris, France; 3INSERM US24/CNRS UMS3633, 75015 Paris, France; 4Faculté de Médecine, Paris Descartes, Université de Paris, 75015 Paris, France; 5Bioinformatics, Biostatistics, Epidemiology and Computational Systems, Institut Curie, 75005 Paris, France; loredana.martignetti@institutimagine.org (L.M.); veronique.stoven@mines-paristech.fr (V.S.); 6INSERM U900, 75005 Paris, France; 7CBIO Mines-ParisTech, 75005 Paris, France; 8Laboratoire de Biochimie Générale, Hôpital Necker Enfants Malades, AP-HP Centre Université de Paris, 75015 Paris, France; 9Centre de Référence Maladies Rares, Mucoviscidose et maladies de CFTR, Hôpital Necker Enfants Malades, AP-HP Centre Université de Paris, 75015 Paris, France; michaela.semeraro@aphp.fr; 10Centre d’Investigation Clinique, Hôpital Necker Enfants Malades, AP-HP Centre Université de Paris, 75015 Paris, France; 11Pneumo-Allergologie Pédiatrique, Hôpital Necker Enfants Malades, AP-HP Centre Université de Paris, 75015 Paris, France; 12European Respiratory Network, ERN Lung, 75015 Paris, France

**Keywords:** cystic fibrosis, kidney, cystic fibrosis transmembrane conductance regulator, klotho, proteomic

## Abstract

**Background:** The prevalence of chronic kidney disease is increased in patients with cystic fibrosis (CF). The study of urinary exosomal proteins might provide insight into the pathophysiology of CF kidney disease. **Methods:** Urine samples were collected from 19 CF patients (among those 7 were treated by cystic fibrosis transmembrane conductance regulator (CFTR) modulators), and 8 healthy subjects. Urine exosomal protein content was determined by high resolution mass spectrometry. **Results:** A heatmap of the differentially expressed proteins in urinary exosomes showed a clear separation between control and CF patients. Seventeen proteins were upregulated in CF patients (including epidermal growth factor receptor (EGFR); proteasome subunit beta type-6, transglutaminases, caspase 14) and 118 were downregulated (including glutathione S-transferases, superoxide dismutase, klotho, endosomal sorting complex required for transport, and matrisome proteins). Gene set enrichment analysis revealed 20 gene sets upregulated and 74 downregulated. Treatment with CFTR modulators yielded no significant modification of the proteomic content. These results highlight that CF kidney cells adapt to the CFTR defect by upregulating proteasome activity and that autophagy and endosomal targeting are impaired. Increased expression of EGFR and decreased expression of klotho and matrisome might play a central role in this CF kidney signature by inducing oxidation, inflammation, accelerated senescence, and abnormal tissue repair. **Conclusions:** Our study unravels novel insights into consequences of CFTR dysfunction in the urinary tract, some of which may have clinical and therapeutic implications.

## 1. Introduction

Cystic fibrosis (CF) is a life limiting disease due to mutations in the cystic fibrosis conductance gene regulator (*CFTR*). The most frequent CFTR mutation is p.Phe508del, which impedes correct trafficking of the protein to the plasma membrane [1]. This is associated with a multiorgan disease combining pancreatic insufficiency, chronic infected bronchopathy, and production of a salty sweat [1]. Improved survival of patients leads to the observation of new complications in older patients with CF. In particular, the prevalence of chronic kidney disease (CKD) seems to be at least twice the general population, with a two-fold increase every 10 years of age, affecting up to nearly 20% of patients older than 55 years of age [2]. It has been claimed that this was related to secondary risk factors such as CF-related diabetes, or aminoglycoside exposure. However, as some patients develop CKD even in the absence of diabetes mellitus or nephrotoxic treatment, it has been suggested that the CFTR defect alters cellular pathways in mammalian kidney cells, leading to cell injury and progressive CKD. Indeed, a spectrum of renal dysfunctions is described from birth in CF patients, including kidney stones, nephrocalcinosis, low molecular protein urinary loss, and increased renal clearance of drugs [3,4,5]. However, these abnormalities are encountered only in few patients and this contrasts with the high level of CFTR expression in the kidney [6]. The reason for this apparent discrepancy is unknown.

To get new insights into the pathophysiology of CFTR dysfunction in the kidney, we analyzed urinary exosomes from CF patients. Indeed, those extracellular vesicles (EVs) have been largely investigated in kidney or urothelial diseases because they incorporate cellular RNAs, proteins, and metabolites, and thereby reflect cell biology they stem from [7]. Recent studies suggested that studying the proteomic content of urinary exosomes can help unveil abnormal biological processes and contribute to the diagnosis, prognostic assessment, and management of individuals with renal diseases [8]. 

We hypothesized that urinary exosomal proteins might be differentially expressed, depending on the presence of the CFTR protein in kidney cells and that correction of the F508del-CFTR trafficking might modify this profile. CFTR modulator therapies currently administered to F508del homozygotes CF patients include Orkambi^®^ (composed of lumacaftor, a CFTR trafficking corrector, and ivacaftor, an activity potentiator) and Symdeko^®^ (which includes ivacaftor and tezacaftor, a closely related derivative of lumacaftor) [9]. Both combo-therapies display similar efficiency in patients. We analyzed urinary exosomes of patients with CF and healthy controls based on their protein content determined by high resolution mass spectrometry, in combination with gene set enrichment analysis (GSEA). These results were compared to those obtained in exosomes collected in patients treated with CFTR modulators. 

## 2. Results

### 2.1. Description of Patients

A total of 8 controls (5 to 17 years), and 19 CF patients were enrolled in the study. Diagnosis of CF was supported by an elevated sweat test and presence of 2 CF-causing mutations. Twelve patients (5 to 19 years) were not treated with CFTR modulators and had a mean percentage predicted forced expiratory volume in 1 s (ppFEV_1_) of 98 (7) % (mean (SD)). Seven other patients (14–31 years) had initiated lumacaftor/ivacaftor Orkambi^®^ (*n* = 5) or tezacaftor/ivacaftor Symdeko^®^ (*n* = 2) for at least 1 year and had a mean ppFEV_1_ of 75(8). 

### 2.2. Urine Exosomes Characterization 

The amounts of protein extracted after exosome enrichment were similar in controls (3.5 ± 1.9 µg/µL) and in patients (3.5 ± 2.97 µg/µL). 

During a first set of experiments, urinary extracellular vesicles were isolated by differential centrifugation combined with density gradient fractionation. These nanovesicles were analyzed by TEM (Figure 1). They showed the cup-shaped morphology characteristic of exosomes with a size between 50 to 150 nm. Western blotting showed the presence of specific exosomal markers (CD63, CD81, CD9, TSG101, syntenin-1). Fractionation on a step sucrose gradient identified exosomes in the 200,000× *g* ultracentrifugation pellet as well as in fractions F4 (40 to 50% sucrose) and F5 (50 to 60% sucrose), corresponding to 1.17 to 1.23 g/mL on sucrose gradients (Appendix A). 

Mass spectrometry confirmed the presence of CD9, CD63, and CD81 as well as other proteins involved in exosome biogenesis, such as programmed cell death 6-interacting protein (Alix), programmed cell death protein 6 (PDCD6), and flotillin-2. As there was no main difference in exosomal enrichment between F4 sucrose density fraction and the 200,000× *g* ultracentrifugation crude pellet, all experiments were then performed on the 200,000× *g* ultracentrifugation pellet. 

### 2.3. Proteomic Analysis of Urinary Exosomes between CF Patients and Controls

Mass spectrometry analysis allowed the identification of 2315 proteins across all subjects. Of these, 1107 could be reliably quantified and were considered eligible for statistical analysis (Appendix A for complete list). They showed high abundance of proteins usually found in exosomes irrespective of the genotype, such as tetraspanin and integrin proteins (e.g., CD63, CD9, CD81, CD82), Rab GTPases, annexins, flotillin, molecular chaperones (e.g., heat shock protein HSP90), tumor susceptibility gene 101 protein (TSG101), as well as different endosomal sorting complex proteins. 

Other proteins, characteristic of the kidney and the urothelium, were identified. Those included: Podocin (glomerular podocytes); aquaporin-1 (proximal tubule); prominin 2, klotho (distal tubule); SLC12A1 (descending limb of the loop of Henle); uromodulin (ascending limb of the loop of Henle); aquaporin 2, mucin-1 (collecting duct); uroplakin 1 and 2 (urothelium); P2X receptors (bladder epithelium); and epidermal growth factor receptor widely expressed in renal epithelium.

One hundred and thirty-five proteins were differentially expressed amongst controls and CF patients not treated by CFTR modulators (Appendix A for complete list). Seventeen proteins were upregulated in CF patients and 118 were downregulated, based on the significance level of the t-test (FRD (false discovery rate) < 0.05). 

Heat maps of the differentially expressed proteins showed a clear separation between CF and control patients (Figure 2). The upregulated proteins included (i) epidermal growth factor receptor (EGFR); (ii) transglutaminases (TGM1/3); (iii) prelamin-A/C, precursors of lamin A/C; (iv) proteasome subunit beta type-6 (PSMB6), one of the essential subunits that contributes to the 20S proteasome complex; (v) dermcidin, an antimicrobial peptide generating pro-inflammatory cytokines and chemokines; (vi) caspase 14, a protein involved in apoptosis; (vii) protease inhibitors (serpin B12, cystatin A); and (viii) desmosomal proteins such as desmoglein, or proteins anchoring intermediate filaments to the desmosomes junction (plakoglobin, desmoplakin). 

The downregulated proteins were involved in (i) oxidative stress control such as glutathione S-transferases (GSTM2, GSTM3, and LANCL1), extracellular superoxide dismutase (SOD3), klotho, (ii) phagosome maturation (Ras-related protein Rab-34 and RaB-20), (iii) endosomal sorting complex required for transport (ESCRT) (charged multivesicular body proteins, vacuolar protein-sorting-associated protein, serine/threonine-protein kinase ULK3, IST1 homolog); (iv) epithelial cell adhesion (contactin-1, hemicentin-1), and (v) matrisome, i.e., proteins from the extra cellular matrix (ECM) (such as cadherins and collagen paralogs, heparin sulfate proteoglycan, FRAS1-related extracellular matrix protein, fibulin-1, thrombospondin-1, von Willebrand factor A domain-containing protein 1, basigin).

Treatment with Orkambi^®^ or Symdeko^®^ yielded no significant modification of urinary exosome proteomic content, as shown in the dendrogram of patients’ classification (Appendix A).

### 2.4. Gene Set Enrichment Analysis (GSEA) between CF Patients and Controls

Gene set enrichment analysis was based on the differential proteomic data set of CF patients not treated with CFTR modulators versus healthy controls. 

GSEA revealed 94 gene sets that were significantly dysregulated (FDR < 0.1); 20 of them had positive scores, indicating downregulation in CF, and 74 had negative scores, indicating upregulation in CF (Appendix A for complete list). We compared the consistency of differential protein expression and the top scoring gene sets enriched between CF and healthy controls. The top-ranked upregulated genes belonged to the proteasome gene set, while matrisome and endosomal sorting complex required for transport (ESCRT) gene sets were the most significantly downregulated. Figure 3 reports the GSEA enrichment plots obtained for those genes, together with the visualization of corresponding proteins on volcano plots. 

## 3. Discussion

Our study provides the first characterization of the proteomic content of urinary exosomes from CF patients. It unravels consistent protein expression alterations between CF and healthy controls, and yields novel insights into consequences of CFTR dysfunction in the urinary tract, some of which may have clinical and therapeutic implications. 

### 3.1. Cystic Fibrosis and Kidney Tubular Dysfunction

The main result of our study is the clear difference between CF patients and healthy controls at the proteomic level. We recently showed that respiratory exosomes from bronchoalveolar lavages (BAL) from CF patients contained high levels of proteins involved in oxidative stress and in leukocyte chemotaxis in comparison to those from asthmatic patients [10]. However, it was impossible to decipher whether this was due to an intrinsic effect of the *CFTR* mutation or to the consequences of lung infection. As the urine is sterile, the study of exosomes present in this biofluid undoubtedly provides information for CFTR-related renal disease and possibly CF physiopathology [11]. 

The kidney has been little studied in CF because few patients have a clinically relevant disease [1,2,3,4]. This is in contrast with abundant expression of CFTR in the renal parenchyma mainly at the apical cellular membrane in all nephron segments, from the proximal tubules to the collecting ducts [4,5,12,13,14]. This discrepancy has been related to the high expression in the renal medulla of a CFTR splicing variant, conserving functional characteristics, which could compensate for the functional defects of the CFTR mutation [5,15,16], as well as a high residual expression of F508del-CFTR reaching the apical membrane in the kidney [5].

Subtle tubular dysfunction has nevertheless been evidenced in CF, including decreased reabsorption capacity of glucose, sodium, chloride, calcium, phosphate, uric acid, amino acids, and proteins of low molecular weight [4,17]. Moreover, concern is rising about increasing number of adult patients with chronic kidney disease [2,18]. Our data provide interesting mechanistic insights for these observations.

### 3.2. Dysregulated Proteostasis 

CF exosomes showed increased levels of a proteasomal subunit (proteasome subunit 6 protein), and more generally, proteasome gene set enrichment, indicating upregulation of the proteasomal pathway in the CF urinary tract. Upregulation of proteasome subunits has also been observed in several other CF tissues, including the lung as an adaptation mechanism to recycle the excess of misfolded F508del-CFTR proteins [19]. This is in contrast with downregulation of ESCRT proteins, a set of proteins which drive protein sorting at endosomes by polyubiquitination, an essential process to destroy misfolded proteins, including membrane-bound F508del-CFTR [20].

Downregulation of proteins involved in phagosome maturation, such as RAB-34 and RAB-20 [21], suggest a defective autophagy, as already described in epithelial F508del respiratory cells [22]. Very interestingly, in the respiratory epithelial cell, the proposed mechanism for defective autophagy involves accumulation of transglutaminases (TGMs) which crosslink Beclin-1, an important protein of the autophagosome, and hence decrease the formation of autophagosomes [23,24]. In the present study, the increased expression of TGM1 and TGM3 supports this hypothesis. 

Proteasome degradation, delivery of ubiquitin-tagged proteins to the endosome and autophagy pathways are tightly inter-related, and imbalance in either may lead to the accumulation of misfolded proteins and cellular dysfunction [23]. This exosomal proteomic-evidenced signature suggests that CF kidney cells adapt to the CFTR defect by upregulating proteasome activity but that, concomitantly, autophagy and endosomal targeting is impaired as also observed in respiratory cells. 

Accumulation of abnormal protein should result in tubular cell damage along with time. As proximal tubule dysfunction progresses slowly to tubulo-interstitial injury, these changes would only affect long-term renal function. Indeed, prevalence of chronic kidney disease (CKD) seems to be at least twice the general population in CF patients, with a two-fold increase every 10 years of age, affecting up to nearly 20% of the patients older than 55 years of age [2,18].

### 3.3. Dysregulated Tissue Repair and Potential Pro-Aging CF Disease Signature in Urinary Exosomes 

The most upregulated protein in our exosomal proteomic data set was the epidermal growth factor receptor (EGFR), a tyrosine kinase receptor, whose binding induces various downstream signal transduction, including cell proliferation, inflammatory processes, oxidative process, and extracellular matrix regulation—all of them being involved in the onset and progression of renal damage [25]. Therefore, increased EGFR may trigger kidney damage by a number of different signaling pathways. 

First, decreased urinary exosomal levels of glutathione S-transferases (GSTM2, GSTM3, and LANCL1), and superoxide dismutase (SOD) suggest an imbalance in the anti-oxidant response. These proteins are involved in the detoxification of electrophilic compounds and superoxide O^2−^ and their defect decreases the antioxidative response.

Second, we observed a decreased expression of klotho in urinary exosomes. This is the first observation of a potential klotho deficiency in the CF kidney. Klotho downregulation is known to enhance cell senescence induced by oxidative stress and the resulting apoptosis by decreasing the SOD response [26,27]. As Klotho is involved in the kidney’s ability to defend against renal insults [5], this defect should favor chronic kidney disease by abnormal tissue repair and defective protection against damage. This “accelerated aging” defect is also suggested by the increased level of caspase 14, a protein involved in apoptosis and of prelamin A/C in CF exosome, as accumulation of lamin proteins, are involved in various neurodegenerative diseases [28,29].

Our data also showed a consistent downregulation of proteins and gene sets involved in matrisome in urinary exosomes. This result has been shown in respiratory epithelial cell cultures, based on decreased expression of a significant number of genes related to “ extracellular space” or “connective tissue disorders” [30,31]. The extracellular matrix is a highly dynamic structure that interacts with cells to regulate proliferation, migration, and differentiation for tissue injury repair [32]. Hence, decreased matrisome proteins and gene expression in CF urinary exosomes could indicate an abnormal tissue remodeling in the CF kidney cells.

Altogether the combination of oxidation, defective extracellular matrix, and accelerated senescence suggest that the CF kidney may be more prone to renal abnormal tissue repair, fibrogenesis, and chronic kidney disease progression. This deserves more basic study in renal cells from patients with CF and animal models.

### 3.4. No evidence of Strong Proteomic Modulation by CFTR Modulators 

Treatment with Orkambi^®^ or Symdeko^®^ yielded no significant modification of the urinary exosome proteomic content. Our data contrast with recently published transcriptomic studies which identified 36 genes significantly modified after Orkambi^®^ treatment [33]. Although this number is small at the genome scale, suggesting that these drugs may have only a very limited effect on transcription, the results highlighted modifications of relevant pathways in the context of CFTR, such as normalization of protein synthesis, decreased expression of cell-death genes, and in clinical responders, changes in oxidative phosphorylation, mitochondrial function, eIF2 and IL-17 signaling. This difference may be due to the fact that this CFTR modulator combo therapy is suboptimal to reverse the renal phenotype. Moreover, it cannot be excluded that some patients might have tubulointerstitial injuries due to previous intravenous aminoglycosides treatment that CFTR modulators might not reverse. However, this concerns a single patient in our data set who was treated with intravenous tobramycin on a regular basis, and this should not be the main explanation. None of the patients was diabetic. Importantly, we did not perform a pre/post CFTR modulator study. Our study therefore may not be able to detect dynamic changes that may occur earlier after initiation of the CFTR modulators. Moreover, proteomic studies provide a lower output of elements as compared to transcriptomic profiling. However, it must be underlined that both our study in urinary exosomes and the transcriptomic studies in airway cells indicate a significant upregulation of genes of proteasome and chaperons involved in protein degradation such as endoplasmin [34]. 

## 4. Materials and Methods

### 4.1. Patient Enrollment

All CF patients were recruited from the pediatric pulmonology department in Necker-Enfants Malades Hospital, France, and healthy subjects were enrolled from routine clinics. The study was approved by the Ile de France 2 Ethics Committee, and written informed consent was obtained from each adult, and parent for children <18 years (CPP IDF2: 03/05/2010; ClinicalTrials.gov NCT 02965326, 16 November 2016). 

None of the patients had abnormal creatinine levels. A subset of patients was treated with CFTR modulators (Lumacaftor/Ivacaftor—Orkambi^®^ or Tezacaftor/Ivacaftor—Symdeko^®^; Vertex Pharmaceuticals Inc., Boston, MA, USA) for at least 3 months. Urine samples were collected at second morning void. 

### 4.2. Exosome Isolation 

Exosomes were isolated by differential ultracentrifugation, as previously published [35,36]. Briefly, urine aliquot was centrifuged at 17,000× *g* for 30 min to remove cell debris. Next, the supernatant was centrifuged at 200,000× *g* for 3 h at room temperature. Pellets were then resuspended in isolation buffer (250 mM sucrose, 10 mM triethanolamine pH 7.6) supplemented with 200 mg/mL dithiothreitol (DTT) for 30 min at 37 °C and centrifuged at 200,000× *g* for 1 h at room temperature (XL-70 ultracentrifuge, Beckman–Coulter, Indianapolis, IN, USA). All the compounds were from Sigma–Aldrich, St. Louis, MO. Those extracellular vesicles were further fractionated on a step sucrose gradient (sucrose layers of density ranging from 1.03 to 1.26) [37]. Briefly, the 200-kg fraction urinary EVs were resuspended in a 0.95-M sucrose solution and inserted inside a sucrose step gradient column made of six 2-mL layers of 0.25, 0.6, 0.95, 1.2, 1.65, and 2 M glucose. The gradient was centrifuged at 200,000× *g* for 16 h at 4 °C and 1 mL fractions were collected from the top of the gradient. All fractions, except the first and the last one, were pooled together to collect material equilibrating at the interface of 2 successive sucrose layers for a total of 7 fractions named F1 to F7 from gradient top to bottom. These fractions were diluted in cold PBS and centrifuged at 100,000× *g* at 4 °C for 70 min. The concentration of exosomal proteins was determined using DC (detergent compatible) protein assay (Bio-Rad Laboratories, Hercules, CA, USA) according to the manufacturer’s instructions. Samples were then frozen at −80 °C and stored until further experiments.

### 4.3. Transmission Electron Microscopy 

Transmission electron microscopy (TEM) was performed as previously published [38]. The samples were mixed initially with 4% paraformaldehyde (in PBS, pH 7.4) in a 1:1 ratio. Then, 200-mesh nickel grids were floated on a droplet of the sample for 10 min at room temperature, and finally washed with a PBS droplet followed by a water droplet twice for 5 min. The sample was negatively stained by floating grids on droplets of 1% uranyl acetate for 1 min. After drying, the grids were examined with a JEOL 1011 electron microscope operated at 80 kV. Between 10 and 20 fields were captured for each sample in order to have a good overview of the disparity in diameter of the isolated structures. Acquired images were analyzed using ImageJ 1.47 v.

### 4.4. Immunoblotting

Urinary EVs were lysed in Radioimmunoprecipitation assay (RIPA) buffer supplemented with protease inhibitors for EV protein quantification (23225, Pierce, Rockford, IL, USA) and EV protein content analysis by western blotting. EV lysates were separated on 4–20% tris-glycine gradient gels (criterion precast gel, Bio-Rad, Hercules, CA, USA) and transferred onto nitrocellulose and PVDF (immobilon, EMD Millipore, Billerica, MA, USA) overnight at 4 °C. The membranes were then blocked with 3% bovine serum albumin (BSA) membranes and probed with the following primary antibodies: CD63 (1:250; 15263, Santa Cruz Biotechnology, Dallas, TX, USA), CD9 (1:1000; MM2-57, EMD Millipore, Billerica, MA, USA), CD81 (1:250; 166029, Santa Cruz Biotechnology, Dallas, TX, USA), TSG101 (1:500, 612696, BD Biosciences, San Jose, CA, USA), syntenin-1 (1:1000; 133267, Abcam, Cambridge, MA, USA), uromodulin (1:1000; 19552, Santa Cruz Biotechnology, Dallas, TX, USA), calnexin (1:250; 6465, Santa Cruz Biotechnology, Dallas, TX, USA), detected by fluorescent secondary antibodies (IRDye, LI-COR, Lincoln, NE, USA). Protein bands were detected by the Odyssey imaging system (LI-COR, Lincoln, NE, USA) and processed by Image J (Version 64-bit for Windows, 2017, NIH, Bethesda, MD, USA).

### 4.5. Sample Preparation for LC-MS/MS Analysis

S-Trap^TM^ micro spin column (Protifi, Hutington, NY, USA) digestion was performed on 20 µg of proteins from urinary exosomes in accordance with the manufacturer’s protocol. Briefly, 5% SDS was added to the samples. Proteins were reduced with the addition of tris(2-carboxyethyl)phosphine (TCEP) to a final concentration of 100 mM and alkylated with the addition of iodoacetamide to a final concentration of 50 mM. Aqueous phosphoric acid was added to a final concentration of 1.2%. Colloidal protein particulate was formed with the addition of 6 times the sample volume of S-Trap binding buffer (90% aqueous methanol, 100 mM TEAB, pH 7.1). The mixtures were put on the S-Trap micro 1.7 mL columns and centrifuged at 4000× *g* for 30 s. The columns were washed four times with 150 µL S-Trap binding buffer and centrifuged at 4000× *g* for 30 s with 180 degree rotation of the columns between washes. Samples were digested with 1 µg of trypsin (Promega France, Charbonnières-les-Bains, France) at 37 °C overnight. Peptides were eluted with 40 µL of 50 mM TEAB followed by 40 µL of 0.2% aqueous formic acid and by 35 µL 50% acetonitrile containing 0.2% formic acid. Peptides were finally vacuum-dried down.

Proteins were identified and quantified by nanoscale liquid chromatography coupled to tandem mass spectrometry (NanoLC-MS/MS; Nano-LC: Dionex, Ultimate 3000 RSLCnano System, Thermo Qexactive Plus, Thermo Scientific, Illkirch, France). Samples were resuspended in 20 µL of 10% ACN, 0.1% TFA in HPLC-grade water. For each run, 1 µL was injected in a nanoRSLC-Q Exactive PLUS (RSLC Ultimate 3000) (Thermo Scientific, Waltham, MA, USA). Peptides were loaded onto a µ-precolumn (Acclaim PepMap 100 C18, cartridge, 300 µm i.d.×5 mm, 5 µm) (Thermo Scientific, Waltham, MA, USA), and were separated on a 50-cm reversed-phase liquid chromatographic column (0.075 mm ID, Acclaim PepMap 100, C18, 2 µm) (Thermo Scientific, Waltham, MA, USA). Chromatography solvents were: (A) 0.1% formic acid in water, and (B) 80% acetonitrile, 0.08% formic acid. Peptides were eluted from the column with the following gradient 5 to 40% (120 min), 40 to 80% (1 min). At 121 min, the gradient stayed at 80% for 5 min and, at 127 min, it returned to 5% to re-equilibrate the column for 20 min before the next injection. Two blanks were run between each series to prevent sample carryover. Peptides eluting from the column were analyzed by data-dependent MS/MS, using the top-10 acquisition method. Peptides were fragmented using higher-energy collisional dissociation (HCD). The instrument settings were as follows: Resolution was set to 70,000 for MS scans and 17,500 for the data-dependent MS/MS scans in order to increase speed. The MS AGC (Automatic Gain Control) target was set to 3 × 10^6^ counts with the maximum injection time set to 60 ms, while the MS/MS AGC target was set to 1 × 10^5^ with the maximum injection time set to 60 ms. The MS scan range was from 400 to 2000 m/z. The dynamic exclusion was set to 30 s duration. Three replicates of each sample were analyzed by nanoLC/MS/MS.

Following LC-MS/MS acquisition, the MS files were processed with the MaxQuant software version 1.5.8.3 (Max Planck Institute, Munich, Germany) and searched with the Andromeda search engine against the database of *Homo sapiens* from swissprot 07/2017. To search parent mass and fragment ions, we set an initial mass deviation of 4.5 and 20 ppm, respectively. The minimum peptide length was set to 7 amino acids and strict specificity for trypsin cleavage was required, allowing up to two missed cleavage sites. Carbamidomethylation (Cys) was set as fixed modification, whereas oxidation (Met) and N-term acetylation were set as variable modifications. The false discovery rates (FDRs) at the protein and peptide level were set to 0.01. Scores were calculated in MaxQuant as described previously [39]. The reverse and common contaminants hits were removed from MaxQuant output. Proteins were quantified according to the MaxQuant label-free algorithm using LFQ intensities [40]. Match between runs was not allowed. 

### 4.6. Gene Set Enrichment Analysis

Gene set enrichment analysis (GSEA) was used to assess the enrichment of gene sets from the Molecular Signature Database (http://www.broadinstitute.org/gsea/msigdb/index.jsp) [41]. To run GSEA, we used a priori annotated gene sets of the Molecular Signatures Database (MSigDB v7.0; Broad Institute, Inc., Massachusetts Institute of Technology, and Regents of the University of California, USA) included in canonical pathway databases (CP collection), a catalogue of 2199 gene sets collected from the publicly available databases BIOCARTA, KEGG, PID, and REACTOME. 

### 4.7. Statistics

Statistical and bioinformatic analysis of MS data including heatmaps, profile plots, and clustering were performed with Perseus software (version 1.6.7.0; Max Planck Institute of Biochemistry, Munich, Germany) freely available at www.perseus-framework.org. The LFQ (label-free quantification) data were transformed in log2, and proteins that were identified in all the samples from at least one group (healthy controls, CF patients not treated with CFTR modulators, CF patients treated with CFTR modulators) were retained for statistical tests. To discriminate differential proteins amongst groups, we performed a t-test using FDR ≤ 0.05 and S0 = 0.1 as a constant added to avoid divisions by an extremely small variance estimate. Hierarchical clustering of significantly differentially expressed proteins was performed in Perseus on logarithmic LFQ intensities after z-score normalization, using Euclidean distances and default parameters. Extraction of annotations GO, keywords, and KEGG pathway, as well as the Fisher test, were performed using Perseus. The Fisher exact test *p*-values were corrected using Benjamin–Hochberg and the threshold for FDR was set to 0.05.

GSEA provided an enrichment score (ES) and a p-value to assess whether the expression of the genes in a given gene set were significantly correlated with one of the groups under study. FDR correction for multiple tests was applied to the enrichment p values of the gene sets and a FDR threshold of 0.1 was defined to select the significantly correlated gene sets. With an overlap statistical analysis applied to differentially expressed proteins defined according to a fold-change cut off, GSEA first ranks all genes (corresponding to the proteins identified) in a dataset, and then calculates the ES for each gene set. ES reflects how often members of that gene set occur at the top (e.g., upregulated) or the bottom (e.g., downregulated) of the ranked dataset [41]. The leading-edge genes include those genes that appear in the ranked list before the point at which the enrichment score reaches its maximum. Therefore, this gene subset can be interpreted as the core that accounts for the gene set’s enrichment signal. 

## 5. Conclusions

Although kidney has not attracted the most attention in CF, our data indicate that this organ might suffer from tissue injury, and suggests a risk to evolve to chronic kidney disease upon patients’ ageing. These features might be detected in older patients and become a source of clinical concern now that survival of CF patients is increasing.

This is the first study showing that CF pathobiology modifies urinary exosome protein composition, and therefore generates potential CF exosomal biomarkers. Taken together, our results identified that CFTR dysfunction causes cellular dysregulations in the kidney, sharing some of the hallmarks of CF airways, such as deregulations of proteasome, ESCRT signaling pathway, perturbation of extracellular matrix, and possibly abnormal tissue repair with accelerated senescence. The decreased expression of klotho might play a central role in this CF kidney signature [42].

This snapshot of the CF exosomal urinary proteome signatures also provides a set of proteins with potential utility as biological markers, in the context of evaluation of new therapeutic strategies. Indeed, urinary exosomes are easily accessible by non-invasive means, which is a strong asset in pediatric patients. Although we were not able to highlight proteins differentially expressed between patients treated with CFTR modulators, future studies may be able to refine a modification of this signature by CFTR modulators. 

Thus, as the CF population continues to survive longer, it should be paid more attention to renal function which should be monitored and checked prior to the administration of potentially nephrotoxic drugs. 

In the volcano plot of proteomic data, the Log2 of the RP (*p* value) statistic is plotted against Log2 of fold change. Bold points in the volcano plot correspond to the proteins resulting from the genes belonging to the gene set. The proteins corresponding to the leading genes are depicted in red while the other genes are depicted in dark gray. Proteins outside the vertical dotted lines have more than 2-fold differential expression. 

## Figures and Tables

**Figure 1 ijms-21-06625-f001:**
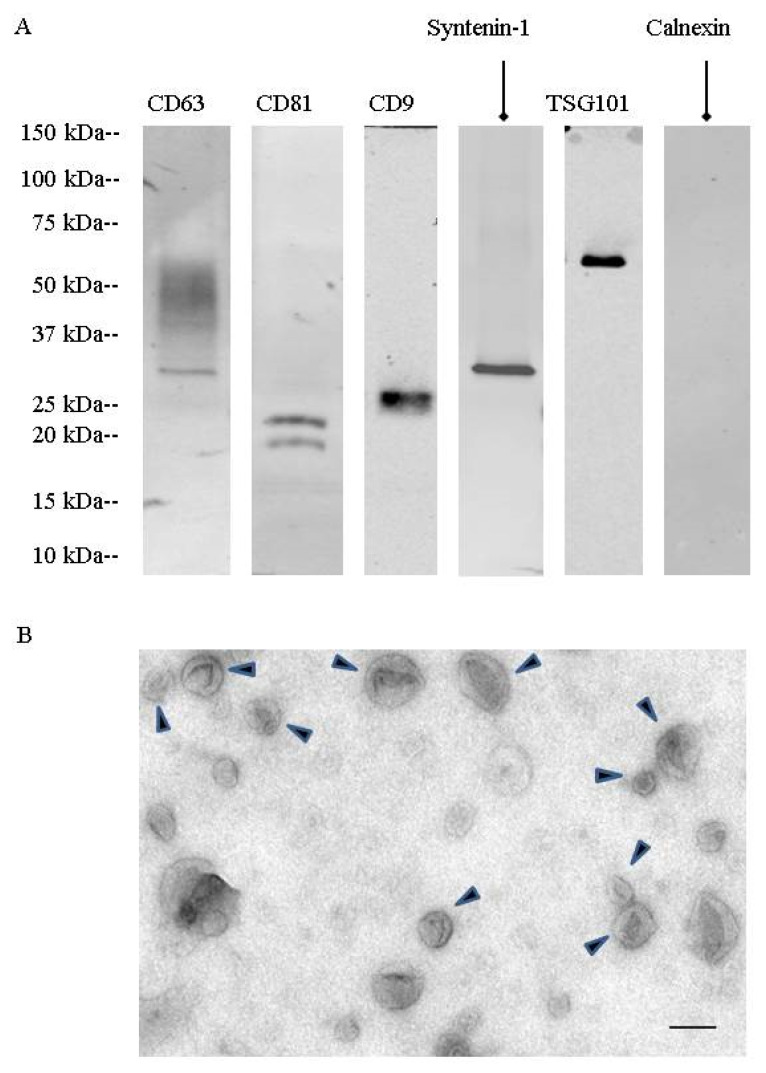
Characterization of urinary exosomes isolated by differential centrifugation. (**A**) Western blotting analysis of 200-kg sample protein contents isolated from urine collected from cystic fibrosis (CF) patients and age-matched control individuals showed the presence of the exosomal markers CD63, CD81, CD9, TSG101, syntenin-1, but not the endoplasmic reticulum-specific marker Calnexin. (**B**) Representative wide field EM image of the contents in the 200-kg pellet showing extracellular vesicles of typical exosome size and shape (scale bar = 200 nm). Arrows: exosomes.

**Figure 2 ijms-21-06625-f002:**
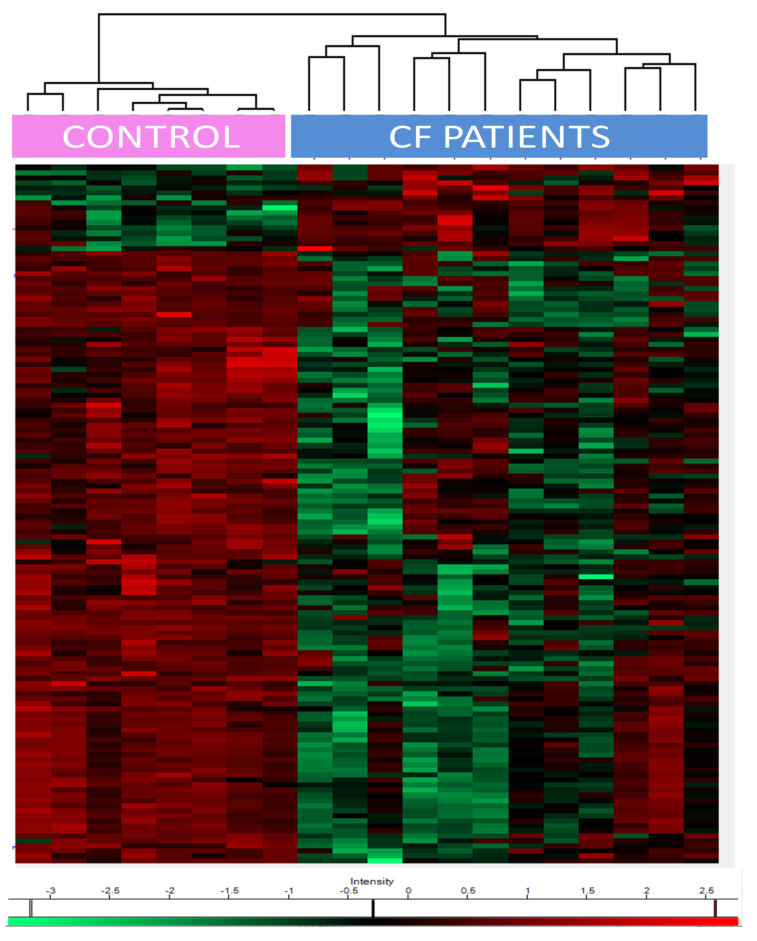
Heatmap of the exosomal proteins differentially expressed between CF patients and healthy controls. Lower and higher values than the mean are represented in green and red scale, respectively.

**Figure 3 ijms-21-06625-f003:**
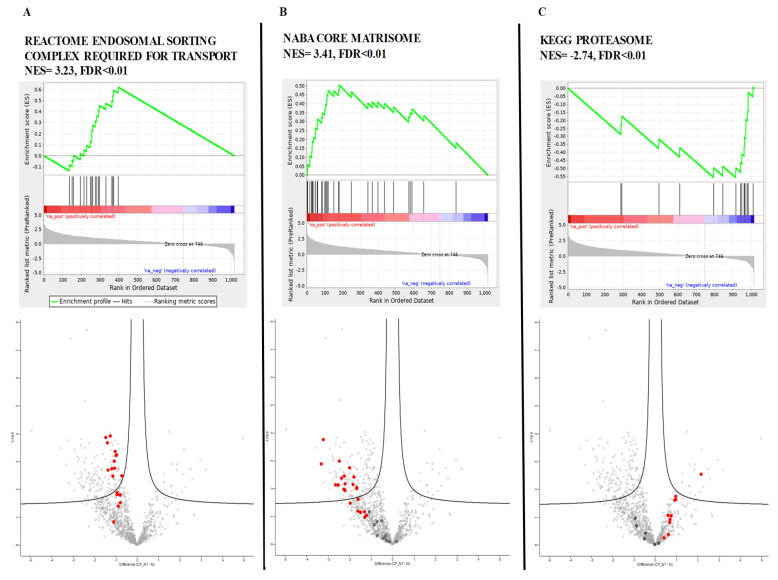
Gene set enrichment analysis (GSEA) enrichment and volcano plots of significantly dysregulated pathways. GSEA enrichment plot and visualization of the corresponding gene set on volcano plot are displayed for (**A**) endosomal sorting complex required for transport (ESCRT), (**B**) matrisome, and (**C**) proteasome pathways. The enrichment score (ES, green line) and the normalized enrichment score (NES) reflect the degree to which the gene set is over-represented at the top or bottom of the ranked list of genes included in the analysis. Black bars illustrate the position of genes belonging to the gene set in the ranked list of genes. In the volcano plot of proteomic data, the Log2 of the RP (*p* value) statistic is plotted against Log2 of fold change. Bold points in the volcano plot correspond to the proteins resulting from the genes belonging to the gene set. The proteins corresponding to the leading genes are depicted in red while the other genes are depicted in dark gray. Proteins outside the vertical dotted lines have more than 2-fold differential expression.

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
