# Peer review of "Urinary Exosomes of Patients with Cystic Fibrosis Unravel CFTR-Related Renal Disease"

_ijms, 2020, doi:10.3390/ijms21186625_

Round 1

Reviewer 1 Report

In this article S. A. Gauthier, I. Pranke, I. Sermet-Gaudelus and co-workers have reported their investigations towards Cystic Fibrosis and CFTR related renal disease. Urine samples were collected from 19 CF patients (among those 7 were treated by CFTR modulators), and 8 healthy subjects. Urine exosomal protein content was then determined by high resolution mass spectrometry. Interestingly, 17 proteins were upregulated in CF patients and 118 were down regulated. Gene Set Enrichment Analysis revealed 20 gene sets upregulated and 74 downregulated. Noteworthy, treatment with CFTR modulators yielded no significant modification of the proteomic content. These results highlight that CF kidney cells may adapt to the CFTR defect by upregulating proteasome activity and that autophagy and endosomal targeting are impaired.

Overall, this investigation is important as it unravels novel insights into consequences of CFTR dysfunction in the urinary tract, some of which having possibly clinical and therapeutic applications.

The article is well written, and the experimental part and results and discussion sections are appropriately detailed and of good scientific quality. To my opinion, it fully meets the requirements of quality and originality for publication in International Journal of Molecular Sciences.

However, minor revisions must be fixed.

Author names (L4-5): please homogenize by writing first names and last names or change Isabelle Sermet-Gaudelus by Sermet-Gaudelus I.

L17: Enfants

L92: 1.23 g/mL

L100 and L103: space between 200 and kg.

Figure 2. For sake of clarity, could it be possible to enlarge heatmap (i.e., landscape format?) and combine with Figure S2 and protein names.

In the text, references must be reported in brackets and with a normal size.

L282, L283: mL and not ml.

L282: 0.25 M, …. (space, no space?)

In the method section and elsewhere in the text, please correct to the good writing styles and conventions in chemistry:

I.e., XX °C (space between X and °C); XX mL (space between X and mL; and mL, L is a cap); XX µL (space between X and µL), XX mM (space between X and mM),

L413-416: Homogenize abbreviations: for instance ISG and IS?.

Reference section: Please correct to the good template. See at www.mdpi.com › files › ijms-template

Author Response

Paris, 4 September 2020

Dear Reviewer,

Thank you very much to accept a revision for our paper “Urinary exosomes of patients with Cystic Fibrosis unravel CFTR related renal disease” by Gauthier, Pranke and Jung, co first authors.

We have revised the manuscript and provide here a point by point response.

“English language and style are fine/minor spell check required”

We thank the reviewer for this remark. We have carefully reviewed the manuscript.

“Author names (L4-5): please homogenize by writing first names and last names or change Isabelle Sermet-Gaudelus by Sermet-Gaudelus I.”

Isabelle Sermet-Gaudelus has been changed by Sermet-Gaudelus I

“L17: Enfants”

This has been corrected.

“L92: 1.23 g/mL”

1.23 g/ml has been changed for 1.23 g/mL.

“L100 and L103: space between 200 and kg.”

Space between 200 and kg has been introduced.

“Figure 2. For sake of clarity, could it be possible to enlarge heatmap (i.e., landscape format?) and combine with Figure S2 and protein names.”

The Heat map has been enlarged as requested. We can not combine the Figure S2 with the heatmap in the main text because it shows the difference between Healthy subjects and non treated CF patients (in the case CFTR modulation might induce proteic changes). The heatmap in the figure S2 shows Healthy versus ALL CF patients (treated and not treated). Therefore we think, for ease of reading, that it is better to leave the figures as in the initial manuscript.

“In the text, references must be reported in brackets and with a normal size.”

This has been corrected.

L282, L283: mL and not ml

This has been corrected.

L282: 0.25 M, …. (space, no space?)

This has been corrected.

In the method section and elsewhere in the text, please correct to the good writing styles and conventions in chemistry:

I.e., XX °C (space between X and °C); XX mL (space between X and mL; and mL, L is a cap); XX µL (space between X and µL), XX mM (space between X and mM),

We apologize for so many mistakes. This has been corrected.

L413-416: Homogenize abbreviations: for instance ISG and IS?.

This has been corrected.

Reference section: Please correct to the good template. See at www.mdpi.com › files › ijms-template

This has been corrected.

Reviewer 2 Report

Very nicely presented manuscript.

My only two comments are:

1. The full name of CFTR has to be given in the abstract also.

2. Is there a correlation between the transcriptosome and exosome genes-proteins affected? (https://doi.org/10.3390/genes11050546). Please discuss.

Author Response

Paris, 4 September 2020

Dear Reviewer,

Thank you very much to accept a revision for our paper “Urinary exosomes of patients with Cystic Fibrosis unravel CFTR related renal disease” by Gauthier, Pranke and Jung, co first authors.

We have revised the manuscript and provide here a point by point response.

“1. The full name of CFTR has to be given in the abstract also.”

We thank for this remark. The full name Cystic Fibrosis Transmembrane conductance Regulator has been introduced in the abstract.

“2. Is there a correlation between the transcriptosome and exosome genes-proteins affected? (https://doi.org/10.3390/genes11050546). Please discuss.”

We thank the reviewer for this interesting question. When comparing  differentially expressed proteins in urinary exosomes detected in our study with results obtained in transcriptomic studies as reported in the above paper, we observe in both, the significant upregulation of genes of proteasome and chaperons involved in protein degradation such as endoplasmin. Our study did not allow to detect the alteration of proteins involved in mitochondrial oxidoreductase activity, lipid metabolism, inflammation and defense that have been observed in transcriptomic studies. However, this is not conclusive, since it can be due to the lower output of proteins by MS technology as compared to transcriptomic profiling.